# Preventing Perinatal Depression: Cultural Adaptation of the Mothers and Babies Course in Kenya and Tanzania

**DOI:** 10.3390/ijerph20196811

**Published:** 2023-09-23

**Authors:** Huynh-Nhu Le, Elena McEwan, Maureen Kapiyo, Fidelis Muthoni, Tobias Opiyo, Kantoniony M. Rabemananjara, Shannon Senefeld, John Hembling

**Affiliations:** 1Department of Psychological and Brain Sciences, The George Washington University, Washington, DC 20006, USA; krabema@gwmail.gwu.edu; 2Catholic Relief Services Headquarters, Baltimore, MD 21201, USA; elena.mcewan@crs.org (E.M.); shannon.senefeld@crs.org (S.S.); john.hembling@crs.org (J.H.); 3Catholic Relief Services, Nairobi 49675, Kenya; akinyikapiyo@yahoo.com (M.K.); fidelis.muthoni@crs.org (F.M.); tobias.opiyo@crs.org (T.O.)

**Keywords:** perinatal depression, Mothers and Babies Course, cultural adaptation, Framework for Reporting Adaptations and Modifications-Enhanced/FRAME, prevention, Kenya, Tanzania

## Abstract

Pregnant women and mothers in sub-Saharan Africa are at high risk for perinatal depression, warranting a need to develop culturally tailored interventions to prevent perinatal depression. This paper documents the process of adapting an evidence-based preventive intervention developed in the United States, the Mothers and Babies Course (MBC), to fit the contexts of rural pregnant women and mothers of young children in Kenya and Tanzania using the updated Framework for Reporting Adaptations and Modifications-Enhanced (FRAME). Data from informant interviews and field observations from the planning and implementation phases were used to make adaptations and modifications of the MBC for perinatal women through the eight aspects of FRAME. Follow-up field visits and reflection meetings with case managers and intervention participants indicated that the adapted version of the MBC was well accepted, but fidelity was limited due to various implementation barriers. The FRAME provided an optimal structure to outline the key adaptations and modifications of a preventive intervention intended to maximize engagement, delivery, and outcomes for high-risk perinatal women in rural settings.

## 1. Introduction

Pregnant and postpartum mothers in sub-Saharan Africa, the majority in low-income and middle-income countries (LMICs), are at higher risk for perinatal depression (PD) than women in high-income countries [1]. PD is among the most common perinatal mental disorders affecting childbearing women in sub-Saharan Africa, with prevalence rates ranging between 10% to 35% for prenatal depression and from 6.9 to 50% for postpartum depression [2]. The high rates of PD in sub-Saharan Africa are attributed to a range of risk factors, which include demographic (poverty, low education, psychological (depression history), interpersonal (intimate partner violence, limited social support, and marital conflicts), and cultural (e.g., failure to adhere to perinatal traditions) [3,4]. Untreated PD is associated with negative consequences for mothers and infants [1,2,5].

The high prevalence and negative consequences of PD warrants prevention efforts for women in sub-Saharan Africa. However, most research has focused on developing treatments for postpartum depression. Evidence-based psychological interventions documented in the literature to treat PPD in LMICs include cognitive-behavioral therapy (CBT), interpersonal psychotherapy, and problem-solving therapy [6]. To our knowledge, the only preventive intervention for perinatal women documented in South Africa is Mamekhaya, which provides psychosocial support for HIV-positive women receiving prevention of mother-to-child transmission of HIV services [7]. The intervention group demonstrated a small effect in reducing depressive symptoms and a moderate effect in increasing knowledge of HIV compared to the usual care group. 

The limited focus on preventive interventions for PD in sub-Saharan Africa warrants additional research in this area. Yet, efforts to systematically document culturally adapted interventions have been limited [8], thus stemming progress in implementation science and cultural adaptation science [9]. To address this limitation, Stirman et al., 2019 [8] refined their original framework for reporting adaptations and modifications to evidence-based interventions: the Framework for Reporting Adaptations and Modifications-Enhanced (FRAME). The purpose of this paper is to document how the Mothers and Babies Course (MBC), a cognitive–behavioral (CBT) intervention for perinatal mothers at risk for depression originally developed in the United States [10], was adapted to fit the contexts of rural pregnant women and mothers of young children in Kenya and Tanzania using the FRAME. As the MBC is integral to the adaptation process, we first describe the history and background of this intervention. 

### The Mothers and Babies Course: History and Background

The MBC was developed as a group intervention to prevent postpartum depression for low-income English- and Spanish-speaking pregnant women at high depression risk in the United States (U.S.; [11]). The MBC has demonstrated efficacy in reducing depressive symptoms and incidences of major depressive episodes among low-income women at high risk for depression in the U.S. [10,11,12]. This intervention was highlighted as one of the evidence-based counseling interventions that was recommended by the U.S. Preventive Services Task Force in 2019 [13].

The MBC is based on CBT principles, an evidence-based treatment and preventive intervention for depression for many populations, including women and families [14,15]. To increase the relevance for low-income minority populations, who often have difficult and uncontrollable life circumstances, the MBC teaches CBT concepts through a reality-management model [12]. The *internal reality* refers to one’s thoughts and values, and the *external reality* refers to one’s behaviors or interactions with others. The MBC teaches participants to manage their mood—by changing their internal reality (thoughts) and external reality (pleasant activities and interpersonal relationships). The MBC also integrates elements of attachment theory, proposing that the quality of a child’s relationship with their primary caregiver during infancy has a sustained effect on that child’s behavior throughout their life [16].

## 2. Materials and Methods

The FRAME is a comprehensive model, developed and updated by Stirman et al., 2019 [8] to document the adaptation and modifications to evidence-based interventions. The FRAME includes eight aspects that describe this process and the reasons for modifying the intervention: 

(1) *When and how in the implementation process the modification was made.* This aspect describes the timing of the implementation process of the modification of the intervention; 

(2) *Whether the modification was planned/proactive (i.e., an adaptation) or unplanned/reactive*. This aspect describes the extent to which the intervention modifications were planned (ideally “proactive”) prior to delivering the intervention, and/or unplanned (i.e., improvised or changed spontaneously) during the implementation phase in response to unanticipated challenges (“reactive”);

(3) *Who determined that the modification should be made*. This aspect identifies the person(s) who participated in the decision-making process for the intervention modifications;

(4) *What is modified.* This aspect describes the types of changes that were made to the intervention, which can occur at various levels: content (materials), context (delivery), training, and evaluation, as well as at the implementation and scale-up phases; 

(5) *At what level of delivery the modification is made*. This aspect identifies the target group(s) that the intervention was modified for, which has implications for ascertaining the potential effectiveness of intervention delivery;

(6) *Type or nature of content-level modifications*. This aspect describes the specific content-related modifications made within a particular setting (e.g., structure, sessions) and includes details regarding whether and how “drifts” occur in the process of delivering the intervention;

(7) *The extent to which the modification is fidelity consistent*. This aspect describes the extent to which the modifications made to the intervention were consistent with the core elements that comprised the intervention (“fidelity-consistent” modifications) compared to modifications that are peripheral to the intervention elements (“fidelity-inconsistent” modifications), which can impact the outcomes and applicability to a particular setting;

(8) *The rationale for the modifications made.* This aspect describes the intent or goal and specific reasons why the intervention modifications were made, including the various factors/levels (sociopolitical, organizational, provider, and recipient) that inform the adaptation process, and the barriers and facilitators of adopting the intervention at a broader level. 

We organized and analyzed our data using the updated FRAME. These data were gathered from field observations and informant interviews when the MBC was adapted to perinatal mothers in Kenya and Tanzania from the years of 2016–2018. (The informant interviews with staff and participants were conducted as part of quality improvement and intervention adaptation. Therefore, IRB was not obtained for these interviews. We only summarize the overall findings and no identifying information of these participants is reported). Figure 1 provides an overview and timeline of the first two phases of FRAME.

## 3. Results

We describe below in detail our process for the first two phases of the study, planning and implementation, which form the core intervention modifications for Kenya and Tanzania; Table 1 provides an overview of this process. Figure 2 provides a conceptual overview of FRAME in the context of the IMBC modifications. Finally, we conclude with plans for the remaining two phases and lessons learned from field implementation (see Table 2).

### 3.1. Planning Phase

#### 3.1.1. Aspect 1: Modifications: When and How

In 2016, the planning phase began as a collaboration between the MBC’s cocreator (developer) and the senior technical advisor (STA) for Catholic Relief Services (CRS), an international humanitarian agency working with populations and organizations to promote sustainable and collaborative human development. The STA led the Thrive II project, which aimed to increase the attainment of age-appropriate developmental milestones for children under two years in Kenya, Tanzania, and Malawi. Thrive II utilizes a modified version of the Care Group Model [17] comprised group sessions and home visits delivered by community health workers (CHWs) and care group volunteers (CGVs), who, in turn, are trained by nurses.

The impetus for the MBC to be added to the Thrive II came from results from CRS’ first project, Thrive I, indicating that primary caregivers of children 0–5 reported high levels of anxiety (64%) and depression (61%) [18]. However, staff were unprepared to respond to maternal mental health needs, and counseling options were often not available in the areas serving perinatal women [19]. 

#### 3.1.2. Aspects 2–6: Detailed Modifications 

*Planned intervention modifications*. Modifications were made collaboratively between the MBC developer’s knowledge of the MBC in research contexts and CRS staff’s knowledge of what works in their rural settings in Kenya and Tanzania. The MBC was modified in content and training in five ways (aspects 2, 4.1). First, these modifications considered the sociocultural contexts of the recipients in Thrive II (aspect 5.1). These women live in communal rural areas with few resources, are financially dependent on husbands and their families, and have low literacy levels. The intervention’s name was revised to “The Mothers, Babies, and Young Children (MBYC) Course” given its focus on the first two years of life. 

Second, the intervention’s format was changed to match the Thrive II project structure and staffing (aspect 5.1). The original MBC materials included participants’ and instructors’ manuals that were delivered weekly for 90 to 120 min per session and booster sessions occurring after the end of the intervention to reinforce materials. The MBYC would be integrated and also take place biweekly to follow to structure of THRIVE II’s ECD interventions. As the original MBC materials were available in 6, 8, or 12 group sessions, CRS staff decided to implement the longest version to maximize learning, because this was a new intervention that focused on mental health. Participants would partake in both ECD and MBYC interventions every two weeks, starting with the MBYC and ending with the ECD sessions with 45 min in each intervention. At the end of the 6 months, women would “graduate” from the MBYC but continue with the remaining ECD course for the last 6 months. The ECD intervention also included a monthly home-visit component led by the CGV, which was also adopted with the MBC. Home visits are an essential part of the ECD intervention in order to check in individually to address barriers to ECD and MBYC practice. Finally, booster sessions were included at 3 and 6 months following the completion of the MBYC to reinforce the main intervention concepts, ensuring that participants are practicing the concepts that they learned in the MBC to manage stress and decrease the risk of PD.

Third, the MBYC materials were carefully reviewed by the STA, program managers, and the developer to address literacy and cultural contexts (aspect 4.1). Translations of materials addressed local dialects where the intervention was delivered, including how “depression” and “stress” were expressed within the region. Additionally, there was a mood scale in the original MBC that asked women at the end of each session to rate their mood on a nine-point scale, from one (worst) to nine (best) mood each day within the past week. As some of the women do not know how to read numbers, these numbers were added with faces that depict facial expressions of emotions corresponding to a five-point and seven-point scale (Figure 3, parts 1–3). Facilitators approved of these changes, and participants were able to understand the revised mood scale. Furthermore, the examples for pleasant activities were changed to fit the rural contexts in Tanzania and Kenya (aspect 6.1). The materials were then distributed at the training for review. 

Fourth, the facilitators differed between the MBYC and ECD courses, but both interventions took place concurrently (personnel, aspect 5.1). The CGVs, supervised by CHWs, delivered the ECD materials. The CGVs and CHWs were already trained in the ECD materials, participated in Thrive I, had a full caseload, and did not have time to administer another curriculum. Case managers were hired and trained to deliver the MBYC course to maximize the fidelity of the delivery by employing staff who had some mental health background. 

The intervention developer provided training for the MBC in August 2016 to CRS program staff and partners from Kenya and Tanzania. The training format was modified to fit the needs of CRS staff by expanding the two-day training to five days (aspects 4.1, 5.1) to provide a detailed introduction to CBT concepts that were new to attendees and included time to practice these skills in the field.

*Unplanned Modifications.* The first training in August 2016 was well received by attendees and led to additional modifications of the intervention materials (aspect 2). Overall, participants understood the CBT concepts and intervention and appreciated especially the relaxation exercises as a way to manage stress. The field practices suggested that the mothers understood these concepts. However, the participants’ and facilitators’ manuals needed to be further adapted to the needs of the rural African context (aspects 3–5). 

The attendees recommended removing participant manuals because of low literacy, and it was not common practice in these programs to provide a manual (aspect 3.2). However, CRS staff recommended continued use of thought that it was important for women to be able to monitor their mood scales, leading to several modifications to the materials (aspects 4.2, 6.2). First, the mood scales and core contents were included in a small spiral booklet with illustrations that would be easier for the women to carry back and forth to sessions. Second, the mood scale ratings changed from five to seven to represent more variations in the reporting of one’s mood, including suggestions to make the facial expressions more contextually relevant (Figure 3, parts 2 and 3). Third, the names of “Violet and Mary”, a cartoon that shows two women’s days and how their days can change depending on what they do or think were changed to common names in Kenya (Akoth, Akinyi) and Tanzania (Nkwimba, Nchambi) (aspect 6.2). Fourth, the “Take Home Messages” from the original participant manual would be provided to participants at the end of each MBC session to highlight each session’s main concepts. Fifth, the timing of the sessions decreased from the original delivery of 2 h to 45 min to accommodate the corresponding 45 min ECD intervention. Sixth, the biggest change recommended was to include a “flipchart” format to present the intervention materials, which needed to be big enough for the mothers to see while seated in the field. This flipchart includes a picture of the content to be delivered on the front (with a few words capturing the key message), and the instructions and key messages for the facilitators to share on the back. A graphic designer in Kenya developed pictures and materials that are more relevant to the rural East African context, which also aligned well with the existing materials in Thrive II. The facilitator’s manual would retain the “script” version of key messages in the flipchart and other materials not in the flipchart. Thus, the case managers could prepare ahead of time with the facilitators’ manuals and present with flipcharts only in the field. Finally, the intervention’s name was changed back to MBC as the MBYC was too long and the attendees felt that the MBC still represented the intervention as part of Thrive II. All of these changes led to a second modification of the MBC, prior to field implementation. 

Of the original six case managers trained in August 2016, four had a counseling background and two had a bachelor’s degree in a related area—nursing or health education. The original intent was to hire all case managers with specialized mental health experience; however, it was difficult to recruit providers with mental health backgrounds in both countries. In Tanzania, only one of the three case managers (a nurse) stayed on with the project post-August training. Thus, two new case managers were hired, who had a teaching or health background. A subsequent 3-day training was conducted for the two new case managers as a new training and as a refresher course for the initial case managers in both countries. The program managers from Tanzania and Kenya provided this refresher training, which is a reactive change to onboard the new staff in the new intervention (aspect 4). 

#### 3.1.3. Aspect 7: Fidelity 

The core elements of CBT, reality management, and attachment models that comprise the original MBC were retained and are the main theoretical frameworks for the intervention. The three modules of the intervention (pleasant activities, people contacts, and thoughts) remain similar to the original MBC. New checklists were created to assess the fidelity of the MBC in the group sessions and home visits. Home visits were intended to check in with participants who may not have been able to attend some of the interventions in groups. Since the home visits were only part of the original design in the care-group model in Thrive II, not the original MBC intervention, CHWs needed additional training regarding when to refer participants for additional depression risk assessment.

#### 3.1.4. Aspect 8: The Rationale for Modifications

The reasons for modifying the MBC for perinatal mothers living in rural Kenya and Tanzania came from several factors. From a sociopolitical perspective, attention to mental health is limited in rural areas of sub-Saharan Africa, partly due to stigma and to limited resources devoted to mental health [20]. Thus, CRS recognized the importance of addressing mental health for their participants—pregnant women and mothers with young children to maximize their children’s health and development. Second, the MBC modification was made possible from the organizational setting, specifically from CRS which received foundation funding for Thrive II. Given that the new intervention focuses on preventing PD in this population, it was important to train new providers with more “specialized” skills to implement this intervention for the first time in the field. 

### 3.2. Implementation Phase 

#### 3.2.1. Aspect 1. Modifications: When and How 

The implementation phase for the MBC took place from February to July 2017. Three case managers provided the MBC to 1742 (Kenya: *N* = 756; Tanzania: *N* = 986) women who participated in Thrive II. Informed consent was obtained from 1742 women to participate in the MBC. On average, each case manager had between 23 to 27 groups, with 7–10 women per group, which occurred from February to July 2017. In February and March 2017, the intervention developer, the two program managers (one per country), and the technical advisor conducted field visits to observe the sessions being implemented by each case manager. Fidelity checklists were used to aid the observations and provide feedback to case managers on their performance. During this time, each case manager conducted 1–4 MBC sessions. Field observations revealed that case managers differed in how they delivered the intervention content. For example, case managers with teaching or program management backgrounds stood to deliver the content to participants who sat in a circle on the ground; whereas case managers with a counseling background generally used communication strategies for behavioral change (e.g., fostered a group environment that increased group collaboration). These observations led to several modifications thereafter. 

#### 3.2.2. Aspects 2–6: Detailed Modifications 

Two observations were consistent in the field visits in both countries and led to unplanned changes made to the intervention content following the field visits (aspect 2). First, case managers were all using old materials presented in the first training in August 2016. The flipcharts, take-home messages, and spiral notebooks were created after this training, but these materials were not yet available when the groups started due to budget delays. Second, case managers spent little time integrating the reality-management model with the content that they were teaching participants (e.g., thoughts are part of one’s internal reality). This limitation was due in part to time constraints, the lag of time between training and implementation which led to focusing on the more concrete aspects of the intervention, and the lack of integration of the ECD and MBC materials as the content was taught by two different facilitators. 

Although the MBC content did not vary within the sessions observed, case managers differed in what material they chose to emphasize; therefore, the timing of the various activities differed across the same sessions. For example, relaxation was introduced as a coping skill in session 2. The goals were to introduce and define the concept of relaxation, discuss its benefits, demonstrate a breathing exercise in session, and encourage women to practice relaxation between sessions. However, in some groups, the case managers defined this concept but did not demonstrate it in session yet encouraged women to try this on their own, whereas other case managers completed all tasks as described in the facilitator’s manual. 

The intervention developer, STA, and program managers met with the case managers in each country to provide feedback on post field visits and suggested modifications for the implementation phase (aspect 3). These modifications focused less on content and more on the process of delivery—contextual modifications to improve the facilitators’ ability to teach the MBC concepts, which were novel in perinatal mental health in the sub-Saharan rural setting (aspects 4 and 5). Finally, case managers should use the post-training materials as they become available. These modifications considered participants’ limited literacy by using flipcharts and more pictures to teach the MBC intervention (aspects 5, 6). 

#### 3.2.3. Aspect 7: Fidelity

The use of older materials in the first few sessions of the MBC limited the fidelity of the intervention delivery. A review of the fidelity rating forms indicated that the supervisors generally rated the case managers as performing very well and adhering to all content within each intervention session. However, these ratings did not always match what the intervention developer, STA, and program managers observed in the field. Lower ratings suggested “bad” performances, and supervisors did not want to provide lower ratings to the case managers. Instead, the STA and intervention developer reframed the lower ratings as an opportunity to assess areas that needed additional review of the intervention content for participants to apply in their lives. Furthermore, due to budgetary constraints and difficulty in hiring additional case managers, the high number of groups that each case manager was responsible for meant that the case managers had to travel far between villages, making it difficult to deliver the material with optimal fidelity. Therefore, the intervention developer provided additional guidance on how to deliver the content and engage the participants to practice within and outside of group sessions to increase fidelity.

The intervention developer also provided biweekly remote supervision for the case managers to increase the fidelity of the intervention delivery, including guidance on content and process. Based on the fidelity ratings, case managers reported on activities that were successfully implemented, challenging to facilitate, or lacked group participation. Case managers from both countries mostly participated in these supervision calls together, enabling them to learn from their individual styles of delivering the MBC. 

#### 3.2.4. Aspect 8: The Rationale for Modifications

Modifications were made in response to the observations made in the field visits. case managers (providers) were encouraged to use the new materials. From an organizational perspective, this meant that CRS needed to devote additional financial resources to the community to get these materials in a timely manner. The goal was to maximize fidelity to the MBC, thereby increasing retention and improving outcomes for perinatal women living in rural settings in Kenya and Tanzania (recipients), who were also receiving ECD concepts aimed at improving the lives of these mothers and their babies in the first two years of life.

#### 3.2.5. Field Visits during Implementation 

The purpose of the field visit was to elicit feedback from mothers and case managers regarding their experiences with receiving and implementing the MBC, respectively (Table 2). Based on this qualitative feedback, we planned for the next phase of modification, as outlined in FRAME: the scaling-up phase of the MBC intervention. Evaluation data were collected during the implementation phase and will be reported separately. 

In the reflection meeting with five mothers and CGVs, the recipients reported the intervention as very useful and important to them; the most helpful activities increased their awareness of mood and the impacts of stress and depression on their daily lives and relationships. They also liked relaxation and learning ways to manage their daily stressors and relationships with others. They valued the group support, and some women expressed that this was the first time that they had a group that focused on their own wellbeing. They also liked the incentives (e.g., graduation certificates) provided at the graduation of the MBC. Regarding challenges, the women mentioned that the case managers were sometimes late getting the session and/or had to rush off after a session to facilitate another session. Women also wanted monetary incentives for their participation in the intervention (which was not part of the Thrive II program). 

The reflection meeting with the six case managers indicated that they concurred with the mothers’ feedback on the intervention. The case managers appreciated the remote supervision to reinforce the key messages in the session and ways to improve facilitation (e.g., role plays). The case managers reported having difficulty using old intervention materials, and that the new materials came “too late”. Furthermore, participants had varying levels of comprehension of the intervention due to different literacy levels and/or missing previous sessions. Thus, the time allotted for each session was sometimes not enough, particularly if the concepts were more difficult to understand. Specifically, the materials from the thoughts module were more difficult for case managers to facilitate and for participants to understand. Finally, the contents of the MBC and ECD interventions were presented separately by the case manager and CGV, respectively. As a result, the contents of the two interventions were not coordinated, yet there were some opportunities to integrate these interventions. For example, the ECD curriculum stressed the importance of mothers playing and interacting with the baby to promote positive parenting. As depression can impede this interaction, this would have been an ideal opportunity to reinforce the confluence of these key messages. A final challenge was that the case managers were responsible for a large caseload, which led to difficulties with overscheduling, being on time, and with few opportunities to conduct home visits. 

## 4. Discussion

The FRAME was used to document the process and the reasons for culturally adapting the MBC to the contexts of perinatal mothers in Kenya and Tanzania. Through a review and analysis of field notes, informant feedback, and multiple collaborations among the intervention developer, program managers, case managers, and participants, the MBC underwent a series of proactive and reactive modifications during its planning and implementation phases. Follow-up field visits and reflection meetings with case managers and intervention participants indicated that the adapted version of the MBC was well accepted; however, fidelity was limited due to various implementation barriers. 

The adapted MBC was overall feasible and acceptable as an intervention to address mental health for mothers during the critical early childhood period in Kenya and Tanzania. Our work in adapting and modifying the MBC in Kenya and Tanzania is consistent with previous work that has systematically documented the cultural and contextual adaptations of the MBC with different populations (e.g., Central American immigrant mothers in the United States [20], Tribal American mothers [21], and fathers [12,22]). Collectively, this body of work has found that the process of adaptation is culturally humble, iterative, and has the potential for adoption from the communities impacted. 

Our work led CRS to consider how to scale up the MBC to other programs and countries serving perinatal caregivers. To scale up the MBC, CRS needed to address two challenges encountered in the implementation phase. First, it was necessary to integrate the materials relevant from the ECD to the MBC so that these curricula were not perceived as two different interventions. Second, it was important to assess whether the intervention could be delivered by non-mental health specialists given the difficulty of recruiting mental health professionals to work in rural areas. 

The first challenge led to the creation of an “integrated” version of the MBC, which included the identification of the core ECD content (e.g., breastfeeding) that could be affected by PD. This process involved two days of close review of both the ECD and MBC curricula by the case managers and CRS program staff who were most knowledgeable of these interventions. The intervention developer and STA organized the workgroups to review and identify activities in the ECD curriculum that could be incorporated into the three modules of the MBC. Additionally, given the difficulties with comprehension of thoughts, the next iteration added one more session (three to four sessions) for this module, resulting in 13 sessions. A new version, called the Integrated Mothers and Babies Course (I-MBC), was created and will be used as part of a follow-up project to improve the wellbeing of pregnant women and mothers and children < 2 years old.

The second challenge, how to scale up the I-MBC, given the difficulty in recruiting counselors to work in rural areas, was addressed in two ways. First, lessons learned in implementing the MBC indicated that certain activities were more difficult for case managers to deliver with fidelity to participants, including the reality-management model and the cognitive restructuring activities in the thoughts module. The CRS staff and the intervention developer identified these activities, discussed the best ways to deliver them, and provided a role-play scenario for each activity. CRS provided financial resources to a media company to record these activities to serve as “job aids” for future facilitators. Second, CRS staff recommended that future facilitators of the I-MBC include CHWs, a model that is widely used in Africa to address the shortage of mental health specialists via task-shifting [23]. To maximize fidelity, the case managers would become the supervisors to the CHWs in the next project, and the intervention developer would provide supervision for the case managers. To scale up and increase the sustainability of the I-MBC, our lessons learned from the planning and implementation phases suggest that mental health intervention must be embedded within an existing program that promotes the wellbeing of young children. CRS will work with national associations of sister congregations in Kenya, Malawi, and Zambia to support children under two to attain age-appropriate developmental milestones through the Strengthening the Capacity of Women Religious in ECD Project II (SCORE II). 

Our process of cultural and contextual adaptation has limitations, namely that this is a unique process that involves the efforts of multiple stakeholders in two different countries. Although we were successful in documenting this process, our findings may not be applicable to other countries in sub-Saharan Africa, which may have diverse expressions and experiences of PD. We encourage others to follow this process of adaptation and documentation to ensure that the process fits their contexts and settings. 

## 5. Conclusions

Perinatal depression (PD) is a significant public health issue for pregnant women and mothers with young children in sub-Saharan Africa, and interventions are needed to prevent PD in this high-risk group. The FRAME provided a valuable way to document how the MBC, a preventive intervention for PD originally developed for low-income urban women in the U.S., was adapted to fit the contexts of rural pregnant women and mothers of young children in Kenya and Tanzania via integration with an ECD project. This process of adapting, modifying, and implementing the MBC is dynamic, complex, multifaceted, and imperfect with unanticipated challenges along the way, involving multiple stakeholders and their expertise. This process is a worthwhile investment for organizations to make in order to yield successful engagement, delivery, and outcomes for high-risk perinatal populations as we move to scale up and sustainment. This process may help future researchers and providers to document their own process of adaptation and modification of evidence-based interventions to maximize the impact of such interventions in the field to improve the wellbeing and development of mothers and their children. 

## Figures and Tables

**Figure 1 ijerph-20-06811-f001:**
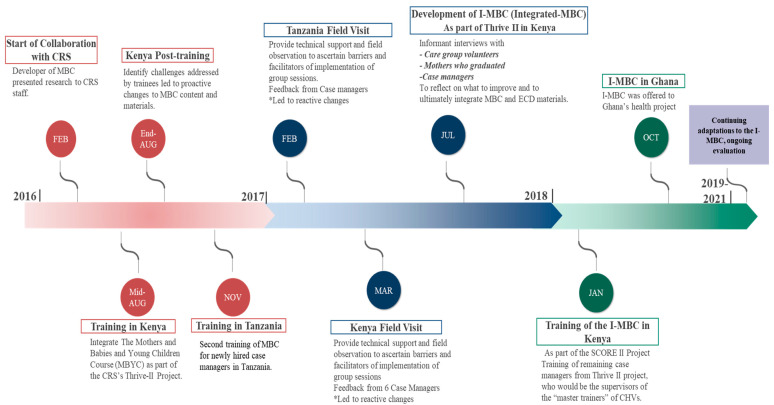
Timeline of the Adaptation of the Mothers and Babies Course in Kenya and Tanzania (2016–2021). Notes: CRS: Catholic Relief Services; MBC: Mothers and Babies Course; MBYC: Mothers and Babies and Young Children Course; I-MBC: Integrated Mothers and Babies Course. * reactive changes refer to changes due to unanticipated challenges.

**Figure 2 ijerph-20-06811-f002:**
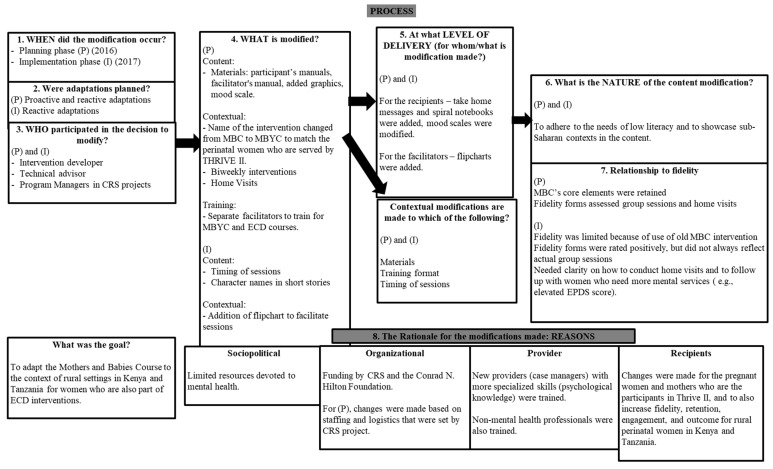
The Mothers and Babies Course based on the updated Framework for Reporting Adaptations and Modifications.

**Figure 3 ijerph-20-06811-f003:**
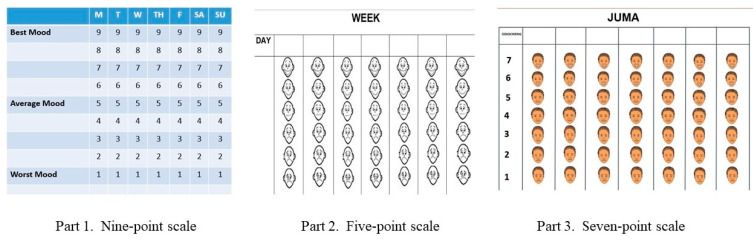
Quick Mood Scale adapted Over Time.

**Table 1 ijerph-20-06811-t001:** Results of the Planning and Implementation Phases for the IMBC.

FRAME Aspects	Planning Phase	Implementation Phase
1. When and how in the implementation process; the modification was made	Collaboration between the intervention developer and senior technical advisor from CRS started in 2016 to add MBC to Thrive II to respond to maternal mental health needs and counseling. The intervention followed the Care Group Model as set by the ECD projects as well as home visits to check in monthly with participants.	Field visits by the intervention developer, two program managers, and the senior technical advisor were done to observe case managers implementing the intervention in Tanzania (February) and Kenya (March), 2017. Changes to the implementation process occurred thereafter.
2. Whether the modification was planned/proactive or unplanned/reactive	Both proactive and reactive	Unplanned/reactive changes in the delivery of the intervention.
**Planned Modifications**		
3.1 Who determined the modification	Intervention developer, senior technical advisor, project coordinators, and project managers in CRS projects (Thrive I and II).	
4.1 What is modified	(a) Content and training changed to adapt to the contexts of perinatal women in Thrive II. (b) The name of the intervention was changed from MBC to MBYC. (c) Format of the intervention changed from weekly to biweekly (d) Materials: Participants’ manuals changed to flip charts and spiral notebooks; the mood scale was changed to have faces instead of numbers, validated with mothers. (e) Translation of all materials to address local dialects/idioms (f) Separate facilitators for MBYC and ECD courses. (g) CRS recommends integrating the MBC into the existing home-visit structure.	
5.1 At what level of delivery the modification is made	(a) Format of training increased from 2 to 5 days to adapt to the needs of local staff. (b) Sessions in the format were changed to adhere to the ECD intervention structure, timing (45 min), and staffing (biweekly). (c) MBYC and ECD took place concurrently. (d) Booster sessions for MBYC occurred at months 9 and 12. (e) Home visits were conducted for both ECD and MBYC interventions.	
6.1 Type or nature content-level changes	Materials were modified to adapt to low literacy levels for participants and rural sub-Saharan contexts.	
**Unplanned Modifications**		
3.2 Who determined the modification	Intervention developer, CRS staff, project coordinator, project managers, and attendees at training.	The intervention developer, senior technical advisor, and program managers provided additional guidance on the delivery/process of intervention based on field visit observations.
4.2 What is modified	(a) Materials: Spiral booklets to show core contents, take-home messages, and mood scales. (b) Add contextualized pictures and activities. (c) Timing of sessions (d) Adding a flipchart in addition to the materials for the facilitators (e) Onboarding of new staff to accommodate refresher trainings.	There was variability in how the intervention content was delivered/how much time was spent on each topic/teaching method in both countries. Therefore, contextual modifications (e.g., location and timing) were made in the delivery of the intervention and ways to improve the participant engagement process.
5.2 At what level of delivery the modification is made	(a) Materials were modified for participants (spiral notebooks). (b) Flipcharts were added to accommodate delivery norms, which aligned with the materials used in the ECD intervention.	Rural Africa context (see planning phase).
6.2 Type or nature content-level changes	(a) Names of characters changed to reflect more common names in Kenya and Tanzania (b) Timing of sessions drifted/changed based on the ECD intervention that is being delivered at each site	Few modifications were made to the content.
7. The Relationship to fidelity	The core elements of the original MBC were retained: CBT, reality management, and attachment models. Fidelity was assessed through different forms for group sessions and home visits. These forms were reviewed by the intervention developer.	(a) Use of the previous version of MBC intervention materials limited fidelity at the beginning of the implementation phase. (b) Fidelity forms of group sessions were rated overall positively but did not always reflect the actual facilitator’s performance. (c) Due to limited budgetary issues, Case managers were responsible for too many groups, decreasing their ability to complete all content within sessions and rushing to deliver the next sessions with limited fidelity. (d) Additional clarity regarding how to conduct home visits and follow up of women who may need additional psychological services.
8. The rationale for the modifications made	(a) Sociopolitical: limited resources devoted to mental health in sub-Saharan Africa (b) Organizational: CRS funding by Conrad N. Hilton to implement Thrive II activities; changes were made based on staffing and logistics set by CRS. (c) Provider: Nonmental health professionals were trained, due to limited providers who have more specialized skills in psychology or mental health, and limited financial incentives to recruit and retain specialized staff in rural areas. (d) Recipient: Adapting it to the needs of the local perinatal women live during the session.	In response to field observations:(a) Organizational: Reactive modifications resulted in CRS devoting additional resources to each community on time and according to their unique needs. (b) Provider: Case managers used new materials created. (c) Recipient: Changes made to increase fidelity, retention, engagement, and outcomes for perinatal women in Kenya and Tanzania.

Notes: CRS: Catholic Relief Services; MBC: Mothers and Babies Course; MBYC: Mothers and Babies and Young Children Course; I-MBC: Integrated Mothers and Babies Course.

**Table 2 ijerph-20-06811-t002:** Summary of Implementation Feedback during Field Visits.

	Implementation Successes	Implementation Challenges
Mothers’ and Careggroup Volunteers’ Feedback (n = 5)	Increased awareness of mood and impact of stressors and depression	Timing of session (e.g., facilitators late to session)
	Enjoyed relaxation and learning activities to manage daily stressors	Expectation of monetary incentives
	Valued group support	
	Enjoyed incentives (graduate certificates)	
Case Managers’ Feedback (n = 6)	Found remote supervision helpful	Used old intervention materials
	Role plays were helpful in reinforcing concepts and improving facilitation	Participants had varying levels of comprehension of materials
		Limited timing of session (e.g., not enough time to cover challenging concepts
		Limited coordination of materials for MBC and ECD contents as they were presented separately
		Large caseloads, leading to overscheduling and timing issues

Note: MBC = Mothers and Babies; ECD = early childhood development.

## Data Availability

The data presented in this study are not publicly available due to the original intent of the data, which were used for quality improvement and intervention adaptation.

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
