# Peer review of "Preventing Perinatal Depression: Cultural Adaptation of the Mothers and Babies Course in Kenya and Tanzania"

_ijerph, 2023, doi:10.3390/ijerph20196811_

Round 1

Reviewer 1 Report

Thank you for the opportunity to review this manuscript, dealing with a very important, yet still overlooked topic. Please find below my suggestions for improving the clarity and flow of the manuscript.

Methods and Results:
I was missing the rationale for including Table 1 in the Methods part of the manuscript- please provide a more detailed explanation of the reason to include Table 1 in Methods, or consider moving this Table to the Results section of the manuscript. In my opinion, the Methods part of the manuscript should be referring to the general description of the method used specifically in the manuscript, Table 1 seems to include also the results.
Also, Table 1 is quite hard to read, some modifications are needed, for instance, dividing the Table into separate sections
On the other hand, Figure 1 with the timeline of the project seems to be more appropriate for the Methods section of the manuscript, than Results.

The results section would benefit from a shortened, briefer, and more concise description of the project aspect, for instance in paragraphs referring to unplanned modifications or home visits.

Conclusion
The conclusion of the manuscript seems very general, referring to general advantages and disadvantages of the FRAME method, it would be beneficial to include more specific conclusions referring to implementing the program, for instance problems with the fidelity fo the materials

Author Response

Response to Reviewer 1

We thank the reviewer for their helpful suggestions to this manuscript, which we believe has strengthened the quality of the manuscript.

Reviewer 1:

Thank you for the opportunity to review this manuscript, dealing with a very important, yet still overlooked topic. Please find below my suggestions for improving the clarity and flow of the manuscript.

Methods and Results:
I was missing the rationale for including Table 1 in the Methods part of the manuscript- please provide a more detailed explanation of the reason to include Table 1 in Methods, or consider moving this Table to the Results section of the manuscript. In my opinion, the Methods part of the manuscript should be referring to the general description of the method used specifically in the manuscript, Table 1 seems to include also the results.

Response:  We have now provided an overview of the 8 aspects and their definitions in the methods section (revised lines 83 to 110).  Table 1 has been reformatted, renamed, and now appears in the results section (p. 5-7).

Also, Table 1 is quite hard to read, some modifications are needed, for instance, dividing the Table into separate sections
On the other hand, Figure 1 with the timeline of the project seems to be more appropriate for the Methods section of the manuscript, than Results. 

Response: Table 1 has been reformatted and now appears in the results section (p. 5-7).  Figure 1 has now been moved to the end of the Methods section (p. 3), which we agree is more appropriate for the Methods section. 

The results section would benefit from a shortened, briefer, and more concise description of the project aspect, for instance in paragraphs referring to unplanned modifications or home visits.

Response: We reviewed the detailed descriptions in the results section and substantially edited these sections, cutting down unnecessary materials. The revised Table 1 serves to provide an overview of these descriptions (p. 5-7).   

Conclusion
The conclusion of the manuscript seems very general, referring to general advantages and disadvantages of the FRAME method, it would be beneficial to include more specific conclusions referring to implementing the program, for instance problems with the fidelity fo the materials.

Response: We believe that this reviewer’s comment regarding the challenges of adapting the MBC was already addressed in the discussion section above the Conclusion section.  We also believe that the conclusion section should be more general with larger implications for the field. To address the reviewer’s suggestion, we added a first sentence in the conclusions section to describe the importance of perinatal depression for perinatal women in sub-Saharan Africa: “Perinatal depression (PD) is a significant public health issue for pregnant women and mothers with young children in sub-Saharan Africa and interventions are needed to prevent PD in this at risk group.” (revised lines 517-519).  In addition, we note that this process of adaptation and modification is “imperfect with unanticipated challenges along the way” (line 524). 

Other minor changes made: We have added and updated the citations and numbering of the references.

Reviewer 2 Report

â—‹ Thanks for your effort in revising your manuscript.

â—‹ This study is significant in that it guides the direction for future research by presenting in detail the process of applying appropriate evidence-based interventions to high-risk perinatal women in rural settings in Kenya and Tanzania using the FRAME framework. However, the following minor revision is required.

â—‹ Line 27~28: “Pregnant and postpartum mothers in sub-Saharan Africa, are at higher risk for perinatal depression (PD) than women in high income countries.” In addition to low and middle income countries, please provide specific reasons including literature on why the risk of perinatal depression (PD) is high.

â—‹ Line 148~155: Please describe the response of the subject and provider to the use of the mood scale.

â—‹ Discussion: The content of the discussion is focused on the research process. Please compare and discuss with the results of other studies that applied the Mothers and Babies Course (MBC).

â—‹ Please suggest the limitations of the study and suggestions for supplementing them.

â—‹ Line 331~ : It would be better for readers to understand if the results of follow-up field visits and reflection meetings with case managers and intervention participants were presented as a table.

Author Response

Response to Reviewer 2

We thank the reviewer for their helpful suggestions to this manuscript, which we believe has strengthened the quality of the manuscript.

Reviewer 2:

Thanks for your effort in revising your manuscript.

â—‹ This study is significant in that it guides the direction for future research by presenting in detail the process of applying appropriate evidence-based interventions to high-risk perinatal women in rural settings in Kenya and Tanzania using the FRAME framework. However, the following minor revision is required.

â—‹ Line 27~28: “Pregnant and postpartum mothers in sub-Saharan Africa, are at higher risk for perinatal depression (PD) than women in high income countries.” In addition to low and middle income countries, please provide specific reasons including literature on why the risk of perinatal depression (PD) is high.

Response: We have now added the prevalence rates for PD and have rearranged the introduction so the risk factors are directly following why the risk is high in the region (revised lines 29-35).

â—‹ Line 148~155: Please describe the response of the subject and provider to the use of the mood scale.

Response: The sentence “Facilitators approved of these changes, and participants were able to understand the revised mood scale” (revised lines 229-230).

â—‹ Discussion: The content of the discussion is focused on the research process. Please compare and discuss with the results of other studies that applied the Mothers and Babies Course (MBC).

Response: We include additional citations to note that others have adapted the MBC and general results from these studies. (revised lines 465-468).   

â—‹ Please suggest the limitations of the study and suggestions for supplementing them.

Response: we note that the main limitation is the unique process of documenting adaptation, which may limit the generalizability in other countries and encourage others to systematically document this process (revised lines 508-513). 

â—‹ Line 331~ : It would be better for readers to understand if the results of follow-up field visits and reflection meetings with case managers and intervention participants were presented as a table.

Response: We added a new table (Table 2) to summarize the results from follow-up field visits (p. 14, revised lines 430-453). 

Other minor changes made: We have added and updated the citations and numbering of the references.

Reviewer 3 Report

Comments and suggestions

It was my pleasure to review this manuscript, which focuses on preventing perinatal depression through the cultural adaptation of the Mothers and Babies Course in Kenya and Tanzania. The manuscript documents how the Mothers and Babies Course (MBC), originally developed as a cognitive-behavioral (CBT) intervention for perinatal mothers at risk for depression in the United States, was adapted to suit the contexts of rural pregnant women and mothers of young children in Kenya and Tanzania. The authors utilized the updated Framework for Reporting Adaptations and Modifications-Enhanced (FRAME) and identified various implementation barriers. Informant interviews and field observations from the planning and implementation phases were used to guide adaptations and modifications of the MBC for perinatal women, following the eight aspects of FRAME.

In summary, this manuscript provides a highly detailed description of the adaptation process. I found the topic to be quite interesting. However, with the sole objective of improving the quality of the manuscript, I would like to offer a few comments.

1.      Introduction:

The introduction provided a good description that emphasizes the importance of this topic, highlighting that untreated perinatal depression is associated with negative consequences for mothers and infants. Therefore, the implementation of a mothers and babies course intervention can prevent and enhance women's mental health, improving their quality of life. However, the introduction needs to include information about the prevalence of perinatal depression in South Africa to help readers understand the importance and necessity of implementing such an intervention.

2.      The Mothers and Babies Course: History and Background (page 2)

The manuscript provides a brief description introducing the Mothers and Babies Course (MBC). Based on these explanations, it establishes the need for conducting a similar study and provides a more precise explanation of the innovative rationale and objectives behind it. Additionally, this section suggests adding past intervention results regarding low-income minority populations, which adds credibility and relevance to the study.

3.      Materials and Methods

The authors mentioned that informant interviews with staff and participants were conducted as part of quality improvement and intervention adaptation. Therefore, IRB approval was not obtained for these interviews. However, it is important to clarify in the Materials and Methods section whether informed consent was obtained during the implementation phase of the MBC from February to July 2017. This phase involved three case managers providing the intervention to 1,742 women who participated in Thrive II.

How to recruit 1,742 women in this study? It is necessary to state in the Materials and Methods section.

4.      Table 1: The FRAME: The Planning and Implementation Phases of the Mothers and Babies Course

The contents of Table 1 contain several abbreviations, such as CRS, ECD, and MBYC. Without explanations for these abbreviations, it becomes challenging to understand their meanings in this section. The same situation is also present in Figure 1 and Figure 2. These abbreviations are described in a later paragraph on page 7.

5. Results

This section of the manuscript provides a well-detailed description that helps us understand how to implement the intervention in Kenya and Tanzania. The content of this process may assist future researchers and providers in addressing their own process of adaptation and modification of evidence-based interventions. This will help maximize the impact of such interventions in the field, ultimately improving the well-being and development of mothers and their children.

Author Response

Response to Reviewer 3

We thank the reviewer for their helpful suggestions to this manuscript, which we believe has strengthened the quality of the manuscript.

Reviewer 3:

It was my pleasure to review this manuscript, which focuses on preventing perinatal depression through the cultural adaptation of the Mothers and Babies Course in Kenya and Tanzania. The manuscript documents how the Mothers and Babies Course (MBC), originally developed as a cognitive-behavioral (CBT) intervention for perinatal mothers at risk for depression in the United States, was adapted to suit the contexts of rural pregnant women and mothers of young children in Kenya and Tanzania. The authors utilized the updated Framework for Reporting Adaptations and Modifications-Enhanced (FRAME) and identified various implementation barriers. Informant interviews and field observations from the planning and implementation phases were used to guide adaptations and modifications of the MBC for perinatal women, following the eight aspects of FRAME.

In summary, this manuscript provides a highly detailed description of the adaptation process. I found the topic to be quite interesting. However, with the sole objective of improving the quality of the manuscript, I would like to offer a few comments.

  1. Introduction:

The introduction provided a good description that emphasizes the importance of this topic, highlighting that untreated perinatal depression is associated with negative consequences for mothers and infants. Therefore, the implementation of a mothers and babies course intervention can prevent and enhance women's mental health, improving their quality of life. However, the introduction needs to include information about the prevalence of perinatal depression in South Africa to help readers understand the importance and necessity of implementing such an intervention.

Response: This is added as part of the introduction: “PD is among the most common perinatal mental disorders affecting childbearing women in sub-Saharan Africa, with prevalence rates ranging between 10% to 35% for prenatal depression and from 6.9 to 50% for postpartum depression [3].” (revised lines 29-32).

  1. The Mothers and Babies Course: History and Background (page 2)

The manuscript provides a brief description introducing the Mothers and Babies Course (MBC). Based on these explanations, it establishes the need for conducting a similar study and provides a more precise explanation of the innovative rationale and objectives behind it. Additionally, this section suggests adding past intervention results regarding low-income minority populations, which adds credibility and relevance to the study.

 Response: we thank the reviewer for their positive feedback on this section.

  1. Materials and Methods

The authors mentioned that informant interviews with staff and participants were conducted as part of quality improvement and intervention adaptation. Therefore, IRB approval was not obtained for these interviews. However, it is important to clarify in the Materials and Methods section whether informed consent was obtained during the implementation phase of the MBC from February to July 2017. This phase involved three case managers providing the intervention to 1,742 women who participated in Thrive II.

Response: the 1,742 women have had informed consent to participate in the MBC, as indicated by the following sentence: “Informed consent was obtained from 1,742 women to participate in the MBC” (revised lines 321-322). 

How to recruit 1,742 women in this study? It is necessary to state in the Materials and Methods section.

Response: we decided to not include this information here, as the purpose of this paper was to document the process of adapting the intervention. We plan to write up the evaluation part following this paper.   

  1. Table 1: The FRAME: The Planning and Implementation Phases of the Mothers and Babies Course

The contents of Table 1 contain several abbreviations, such as CRS, ECD, and MBYC. Without explanations for these abbreviations, it becomes challenging to understand their meanings in this section. The same situation is also present in Figure 1 and Figure 2. These abbreviations are described in a later paragraph on page 7.

Response: we have added these abbreviations, as relevant, at the end of each figure and table.

  1. Results

This section of the manuscript provides a well-detailed description that helps us understand how to implement the intervention in Kenya and Tanzania. The content of this process may assist future researchers and providers in addressing their own process of adaptation and modification of evidence-based interventions. This will help maximize the impact of such interventions in the field, ultimately improving the well-being and development of mothers and their children.

 Response: we thank the reviewer for their positive feedback on this section.

Other minor changes made: We have added and updated the citations and numbering of the references.